# Spherical Nucleic Acids as Precision Therapeutics for the Treatment of Cancer—From Bench to Bedside

**DOI:** 10.3390/cancers14071615

**Published:** 2022-03-23

**Authors:** Akanksha S. Mahajan, Alexander H. Stegh

**Affiliations:** 1Ken and Ruth Davee Department of Neurology, The International Institute for Nanotechnology, The Malnati Brain Tumor Institute, Feinberg School of Medicine, The Robert H. Lurie Comprehensive Cancer Center, Northwestern University, Chicago, IL 60611, USA; akankshamahajan2022@u.northwestern.edu; 2Department of Neurological Surgery, The Brain Tumor Center, Washington University School of Medicine, Alvin J. Siteman Comprehensive Cancer Center, St. Louis, MO 63110, USA

**Keywords:** nanotechnology, Spherical Nucleic Acids (SNAs), RNA interference (RNAi), toll-like receptor (TLR) signaling, cancer vaccine development

## Abstract

**Simple Summary:**

Nanotechnology enables new ways to diagnose and treat cancer. Nanomedicines can increase payload concentration at the disease site, reduce toxicity, and enhance therapeutic effects compared to drugs in their “free” form. Spherical Nucleic Acids (SNAs) emerged as a new class of oligonucleotide nanotherapeutics that are currently being explored as gene-regulatory and immunostimulatory architectures to overcome drug resistance and immunosuppression in solid tumors. This review highlights seminal studies that identified SNAs as a powerful platform for gene regulation, the activation of innate immunity, and the development of next-generation cancer vaccines, discusses recent efforts to translate fundamental discovery from the laboratory into the clinic, and provides an outlook on future research aimed at harnessing the full therapeutic potential of the SNA platform.

**Abstract:**

Spherical Nucleic Acids (SNAs) emerged as a new class of nanotherapeutics consisting of a nanoparticle core densely functionalized with a shell of radially oriented synthetic oligonucleotides. The unique three-dimensional architecture of SNAs protects the oligonucleotides from nuclease-mediated degradation, increases oligonucleotide bioavailability, and in the absence of auxiliary transfection agents, enables robust uptake into tumor and immune cells through polyvalent association with cell surface pattern recognition receptors. When composed of gene-regulatory small interfering (si)RNA or immunostimulatory DNA or RNA oligonucleotides, SNAs silence gene expression and induce immune responses superior to those raised by the oligonucleotides in their “free” form. Early phase clinical trials of gene-regulatory siRNA-based SNAs in glioblastoma (NCT03020017) and immunostimulatory Toll-like receptor 9 (TLR9)-agonistic SNAs carrying unmethylated CpG-rich oligonucleotides in solid tumors (NCT03086278) have shown that SNAs represent a safe, brain-penetrant therapy for inhibiting oncogene expression and stimulating immune responses against tumors. This review focuses on the application of SNAs as precision cancer therapeutics, summarizes the findings from first-in-human clinical trials of SNAs in solid tumors, describes the most recent preclinical efforts to rationally design next-generation multimodal SNA architectures, and provides an outlook on future efforts to maximize the anti-neoplastic activity of the SNA platform.

## 1. Introduction

Oligonucleotide-based therapeutics for gene regulation and immunostimulation have merged as a powerful and novel concept of cancer therapy that can address many of the challenges of conventional drug development [1,2,3,4]. The robust delivery of nucleic acid payloads to tumor sites, particularly those of the central nervous system (CNS), remains challenging, as unmodified oligonucleotides are rapidly cleared through degradation by serum RNases and DNases and inefficiently taken up by cells and tissue [5]. While oligonucleotide carriers, such as polymers, polyplexes, or lipids, display specific safety concerns and delivery limitations, several nanotechnological platforms emerged as therapeutic agents for delivering oligonucleotide payloads to tumors. Spherical Nucleic Acids (SNAs) represent a new class of oligonucleotide-based nanotherapeutics that address the challenges of effective drug delivery to tumor elements, including brain malignancies.

Invented by Chad Mirkin and colleagues [6], SNAs are composed of a nanoparticle core functionalized with a radial shell of highly oriented and densely packed oligonucleotides. SNAs represent a new form of matter, as spherical forms of nucleic acids have properties vastly different from those of their linear counterparts. Unlike linear nucleic acids, SNAs enter many cell types in high quantities without an auxiliary, positively charged transfection agent to overcome the negatively charged plasma membrane. Here, SNAs bind to cell surface pattern recognition receptors, particularly class A scavenger receptors (SRs-A), via oligonucleotide G-quadruplexes [7], and enter cells through caveolae-mediated endocytosis [8]. Many different factors impact the SR-A engagement and cell uptake of SNAs, including oligonucleotide sequence-dependent adsorption of serum proteins [9], and the conjugation of SNAs with a polyethylene glycol backfill used to increase colloidal stability [10].

First-generation SNA nanoconstructs were composed of gold nanoparticle templates modified with a densely packed, highly oriented radial shell of DNA oligonucleotides via thiol adsorption. Research over the past decade has shown that SNAs are highly tailorable structures, which can be composed of various types of oligonucleotides and core materials. Types of oligonucleotides include DNA oligonucleotides [6], locked nucleic acids (LNAs) [11], peptide nucleic acids (PNA) [12], RNA oligonucleotides [13,14,15,16], microRNAs (miRNAs) [17,18,19], RNA-DNA hybrids [20], and ribozymes [21]. Core materials used for SNA synthesis are gold (Au) [6], silver (Ag) [22], iron oxide (Fe_3_O_4_) [23,24], quantum dots (CdSe, CdSe/ZnS) [24], platinum [24], silica (SiO_2_) [25], core–shell (Au@SiO_2_) [25], liposomes [26], lipid nanoparticle [27], poly(lactic-co-glycolic acid) (PLGA) [28], micelles [29], protein cores [30], and T8 polyoctahedral silsesquioxane and buckminsterfullerene C60 scaffolds [31].

The ability to fine-tune core material and oligonucleotide identity has been critical for developing SNAs as cancer therapeutics and diagnostic tools, with many SNA architectures commercialized or enrolled in clinical testing. SNA-based Verigene™ (Luminex, Austin, TX, USA) and SmartFlare™ systems (MilliporeSigma, Burlington, NJ, USA) use SNA technology to detect pathogens or disease-specific markers and enable RNA detection and quantification in live cells, respectively, with a degree of sensitivity and selectivity exceeding that of conventional diagnostic tools [32]. In addition, both gene-regulatory and immunostimulatory SNAs have been evaluated in first-in-human clinical trials to treat psoriasis and solid tumors, including glioblastoma, the most prevalent and aggressive form of brain cancer. These studies demonstrate that SNAs are potent therapeutics that, in contrast to linear nucleic acids, have privileged access to different organ systems, e.g., topically to the skin and systemically to the brain [32]. These studies also highlight the importance of structure and spatial presentation of nucleic acid payloads and demonstrate that nucleic acids gain novel properties suited for applications in biology and medicine when restructured on the nanoscale. Figure 1 shows the timeline of preclinical and clinical SNA development, which is detailed below.

## 2. Gene-Regulatory SNAs to Reactivate p53 Tumor Suppression in Glioblastoma

SNA-mediated gene regulation represents a powerful therapeutic approach for treating cancer and other diseases with a defined genetic basis. Here, SNA target specificity is controlled by the sequence of the gene-regulatory oligonucleotide. Different SNA therapeutics can be generated by modifying the oligonucleotide sequence using digital drug design to target a broad spectrum of oncogenes. Early studies of gene-regulatory SNAs, i.e., gold nanoparticles cores functionalized with antisense oligonucleotides or small interfering RNAs (siRNAs), demonstrated potent knockdowns of model targets (Figure 2). These include luciferase [13]; enhanced green fluorescent protein (eGFP) [25]; genes implicated in cancer cell growth and proliferation, such as the receptor tyrosine kinases epidermal growth factor receptor (EGFR) [33]; and long noncoding RNAs (lncRNAs), e.g., the nuclear-retained metastasis-associated lung adenocarcinoma transcript 1 (Malat1), a critical oncogenic lncRNA involved in metastatic spread of several cancers [34]. The molecular mechanism by which siRNA-carrying SNAs silence the gene expression has been elucidated and includes the helicase Dicer directly accessing the siRNA oligonucleotide on gold nanoparticle templates, followed by the release of the siRNA oligonucleotide and its processing through the canonical RNAi pathway [35].

### 2.1. Preclinical Evaluation of Gene-Regulatory SNAs for the Treatment of Glioblastoma

Building upon these studies, Stegh, Mirkin, and colleagues began evaluating SNAs as gene-regulatory modalities for delivering siRNAs and miRNAs to glioblastoma, the most prevalent and aggressive form of brain cancer. Functional genomics investigations have identified numerous genetic events involved in glioblastoma development. An incomplete understanding of how these genetic aberrations influence tumor response to therapy, combined with the lack of effective delivery of drugs across the blood–brain and blood–tumor barriers (BBBs/BTBs) to the central nervous system, have contributed to making glioblastoma one of the most challenging cancers for which to achieve improved treatment outcomes [36]. Although many oncogene and tumor suppressor gene alterations have been identified in glioblastoma, success in using targeted therapies has yet to be realized [37,38]. The team developed an alternative approach for treating glioblastoma based upon the use of siRNA- or miRNA-carrying gene-regulatory SNAs to silence the expression of the atypical Bcl-2 family protein Bcl2-Like12 (Bcl2L12), a proline-rich protein characterized by a *C*-terminal 14 amino acid sequence with significant homology to the BH (Bcl-2 Homology) 2 domain found in several members of the Bcl-2 family [39,40,41,42]. Bcl2L12 expression is increased in glioblastoma through multiple mechanisms, including gene amplification (as seen as a non-focal gain of chromosome 19(q)), transcriptional upregulation in the absence of gene amplification, and induction of the Bcl2L12-targeting miR-182 [18,39,40,41,42]. Tissue microarray and immunohistochemistry analysis of archived glioblastoma tissue validated Bcl2L12 protein expression in most human glioblastoma specimens and revealed low or undetectable levels in cells of glial origin in normal brain surrounding tumor tissue or in low-grade astrocytoma [42]. *Bcl2L12* is a potential prognostic factor, as glioblastoma patients with high-level overexpression of *Bcl2L12* mRNA have shorter progression-free survival than patients with low or underexpression of *Bcl2L12* [43,44,45]. Molecularly, Bcl2L12 forms a complex with wild-type p53, and upon DNA damage, selectively impedes transcription of specific p53 target genes, including p21 and Bax, which are essential regulators of cell cycling and apoptotic response, respectively [39]. Given the low-level expression of Bcl2L12 in the adult brain, its consistently elevated expression in glioblastoma tumors, the profound negative correlation of its expression with patient survival, and ability to inhibit the p53 tumor suppressor, the inhibition of Bcl2L12 expression represents a rational therapeutic strategy to restore wild-type p53 tumor suppression in glioblastoma.

To neutralize Bcl2L12 expression in an established glioma, Jensen et al. [14] and Kouri et al. [18] comprehensively characterized SNAs with gold nanoparticle templates that carry Bcl2L12-targeting siRNA or miR-182 oligonucleotides, respectively (Figure 2). Both siRNA and miRNA-based SNAs robustly penetrated patient-derived glioma initiating cells (GICs) through SR-A engagement and triggered robust Bcl2L12 target knockdown [14,18]. As SRs are highly expressed on glioma and glioma-associated vasculature [46,47], Jensen et al. used an in vitro non-contact co-culture BBB model to investigate whether SNAs, through SR recognition, endocytotic uptake, and transcytosis, cross the BBB/BTB and infiltrate glioblastoma tumor elements. In this model, which uses human primary brain microvascular endothelial cells (huBMEC) separated from primary cortical astrocytes by a semi-permeable filter insert, fluorochrome-labeled SNAs passed through huBMECs and the filter and rapidly entered the astrocytes [14]. Pharmacological inhibition of SR-A using polyinosinic acid (Poly-I) or fucoidan antagonized the BBB-penetrating capacity for SNAs, suggesting that SR-As are required for BBB transcytosis in vitro [14]. In patient-derived and genetically engineered glioblastoma mouse models, intravenously administered SNAs crossed the BBB/BTB and accumulated in intracranial glioblastoma, as demonstrated by IVIS imaging of mice injected with Cy5.5-labeled SNAs by magnetic resonance imaging (MRI) of glioma-bearing mice treated with gadolinium III (Gd^III^)-conjugated SNAs, and by quantification of gold content in resected tumor tissue using Inductively Coupled Plasmaresonance Mass Spectrometry (ICP-MS) [14,18]. Accumulation and pervasive dissemination into extravascular tumor parenchyma translated into robust intratumoral Bcl2L12 protein knockdown, increased intratumoral apoptosis, impaired tumorigenicity as measured by reduced tumor burden, and prolonged survival of glioblastoma PDX-bearing mice [14,18].

A recent study developed an innovative methodology to optimize SNA gene regulatory activity in vivo. Sita et al. [16] used a glioma orthotopic xenograft model stably co-expressing optical reporters for both luciferase and a near-infrared (NIR) fluorescent protein (iRFP670), with the latter fused to the DNA repair protein O6-methylguanine-DNA-methyltransferase (MGMT). Using non-invasive NIR imaging, this model allowed for a quantitative assessment of MGMT target knockdown and normalization of MGMT expression by tumor volume, measured non-invasively by bioluminescence. Using this model system, Sita et al. determined the optimal dosing and treatment schedule of systemically administered MGMT-targeting RNAi-based SNAs to downregulate the expression of tumor-associated MGMT in vivo robustly and persistently. The concept of a dual reporter xenograft model could be applied to the systematic evaluation and optimization of RNAi-based SNAs with different core materials, surface chemistries, and target specificities.

### 2.2. A Phase 0 Clinical Trial of Bcl2L12-Targeting SNAs in Patients with Recurrent Glioblastoma

Building upon these preclinical studies, a first-in-human, window-of-opportunity phase 0 clinical trial of siBcl2L12-carrying SNAs (drug moniker: NU-0129) in patients with recurrent glioblastoma (NCT03020017) was conducted to assess SNA safety, pharmacokinetics, biodistribution, and target engagement [48]. This clinical trial showed that intravenous infusion of SNAs, similar to results obtained in rodents and non-human primates, was safe at the dose administered and not associated with significant treatment-related toxicities. ICP-MS and X-ray fluorescence microscopy of resected glioblastoma patient tissue demonstrated that intravenously administered SNAs reached the patient tumor and accumulated in the cytoplasm of intraparenchymal tumor cells. Quantification of elemental gold concentration across multiple sections from different patients revealed that glioblastoma cells contained on average 9.1 × 10^−3^ fg of Au per μm^3^, with gold also accumulating in tumor-associated macrophages and the tumor-associated neo-vasculature [48]. SNA uptake into glioma cells was associated with a reduction in Bcl2L12 protein expression and induction of active caspase-3 and p53 proteins, suggesting that the uptake of NU-0129 into tumor cells translated into target knockdown and induction of p53 tumor-suppressive activity [48]. These results establish SNAs as a safe, brain-penetrant precision medicine approach for the systemic delivery of siRNA oligonucleotides to intracranial tumor sites.

## 3. SNAs as a Novel Immunotherapeutic Modality for Solid Cancers

Immunotherapy has rapidly become the fourth pillar of cancer treatment, alongside surgery, radiation, and chemotherapy [49]. It has recently achieved clinical success for melanoma and lung cancer. Here, the use of monoclonal antibodies to inhibit checkpoints (e.g., CTLA-4, PD-1/PD-L1) and activate T cell responses highlights the potential of immunotherapeutics to mount an immune response against tumors [50]. However, complete responses from these therapies are typically limited, and disease progression following the initial response is frequent [51]. The lack of broad clinical success with these agents has fueled the development of new immunotherapeutic modalities that seek to activate both innate and adaptative immunity to “turn cold tumors hot,” rather than simply inhibiting checkpoints. Approaches to stimulate anti-tumor immune responses range from cell-based vaccines to adjuvant therapy. Cell-based treatments are limited in scope and efficacy, showing success in specific cancers (e.g., acute myeloid leukemia or lymphoma), but require expensive, labor-intensive processes due to the handling of human cells [52]. Therefore, the development of effective and easy-to-produce immunotherapies to activate the immune system against specific targets has not been fully realized, especially for difficult-to-treat advanced solid tumors. Nanotechnology offers solutions to these limitations, as it provides a way to orchestrate immune responses by simultaneously triggering multiple pathways. SNAs are particularly well-suited to address these challenges, as they are modular structures that contain various locations for the placement of different immunomodulatory cues and can present and release these stimuli in a spatially and temporally controlled manner.

### 3.1. SNAs to Activate TLR Signaling

Immunostimulatory SNAs are well-defined nanostructures that can present multiple immunostimulatory cues. The prototypic immune-stimulatory SNA architecture consists of a liposomal core conjugated with a radial surface shell of synthetic TLR-agonistic oligonucleotides at a high density (Figure 3). TLRs are transmembrane proteins that are important for triggering innate immune responses against invading pathogens and promoting anti-tumor immunity. Within the endosome, TLR3, TLR7, TLR8, and TLR9 sense different nucleic acids derived from viral or bacterial pathogens. TLR7 and TLR8 are activated by guanosine (G)- and uridine (U)-rich single-stranded oligoribonucleotides. TLR9 senses single-stranded oligodeoxynucleotides that contain repeating unmethylated cytosine-guanosine (CpG) sequences, and TLR3 binds to ∼40–50 base pairs (bp) of non-sequence-specific double-stranded RNA (dsRNA) [53,54,55]. SNAs represent ideal and attractive modalities for activating endosomal TLRs, as SNAs undergo SR-A-dependent endocytosis and are enriched within the endosomal compartment. Recent studies demonstrated that SNAs with liposomal cores that carry CpG oligonucleotides activate TLR9 more potently when compared to linear oligonucleotides and show remarkable activity against murine lymphoma due to multivalent, high-affinity binding to TLR9 [56]. Here, SNA core size, surface curvature, anchor chemistry, and type of oligonucleotide represent important design characteristics for optimal TLR-agonistic activity. SNAs with smaller core templates achieved higher TLR9 activation. In contrast, mixed-curvature architectures accumulated along the endosomal membrane and achieved more potent TLR activation than constant-curvature constructs, which aggregated mainly in the center of the endosomes [57,58]. Furthermore, liposomal SNAs made by directly modifying the surface of a liposomal core containing azide-functionalized lipids with dibenzocyclooctyl-terminated oligonucleotides showed superior stability, cell uptake, and TLR9 activation when compared to conventional cholesterol-based structures, suggesting that different oligonucleotide anchoring chemistries impact SNA performance [59]. In addition, liposomal SNAs were co-functionalized with two different types of CpG oligonucleotides, i.e., class A and class B CpG oligonucleotides. Class A oligonucleotides consist of a partial phosphodiester (PO) backbone with an internal CpG-containing palindrome and a partial phosphorothioate (PS) backbone with poly(G)-terminated strands. Class B CpG oligonucleotides contain a PS backbone and a hexamer CpG motif. Previous studies indicated that the combination of class A and class B CpG oligonucleotide antagonize tumor progression synergistically by activating distinct TLR9 signaling pathways [60]. SNAs functionalized with defined stoichiometries of class A and class B CpG oligonucleotides enabled the robust co-delivery of both CpG classes and synergistically triggered co-stimulatory cues on dendritic cells, compared to mixtures of the linear oligonucleotides. These data suggest that the SNA scaffold is ideal for delivering structurally distinct oligonucleotide payloads and enables the design of multi-sequence oligonucleotide-based combination therapies [61].

Like TLR-9 activating constructs, SNAs carrying TLR7/8-activating RNA, which is attached to hydrophobic cholesterol moieties of the liposomal core template, can engage TLR7/8, trigger NF-κΒ activation in primary immune cells, promote pro-inflammatory cytokine production, and induce the expression of co-stimulatory molecules more effectively than cationic lipid-transfected RNA of the same sequence [62]. In addition, Mirkin and colleagues generated liposomal SNAs that can co-activate multiple TLRs. These SNA architectures contain surface-conjugated CpG oligonucleotides and poly(I:C) encapsulated in the liposome core for concomitant TLR9 and TLR3 activation [63]. On molecular levels, the co-activation of TLR9 and TLR3 resulted in the downstream activation of distinct transcriptional programs, the induction of different interferons, and the synchronization of prolonged expression of co-stimulatory and major histocompatibility complex class II molecules on the cell surface [63], all resulting in the synergistic activation of antigen-presenting cells (APCs). Notably, CpG-carrying TLR9-agonistic SNAs have advanced through a randomized, combined, single-ascending-dose and multiple-ascending-dose phase I trial to assess the safety, pharmacokinetics, and pharmacodynamics in healthy objects (NCT03086278). These structures are currently enrolled in a multi-center phase Ib/II clinical trial (NCT03684785) to evaluate the safety, tolerability, pharmacokinetics, pharmacodynamics, and preliminary efficacy of intratumoral SNA injections alone and in combination with intravenous administrations of the checkpoint inhibitors, pembrolizumab or cemiplimab, in patients with Merkel cell carcinoma, cutaneous squamous cell carcinoma, and advanced solid tumors.

### 3.2. SNA-Based Vaccines Carrying Oligonucleotide Adjuvant and Antigenic Peptide

Immunostimulatory SNAs can be co-functionalized with peptide antigens in various structural arrangements to activate APCs and tumor-targeting effector T cells (Figure 3). Here, SNAs are conjugated with TLR9-agonistic CpG oligonucleotides (i.e., the ‘adjuvant’ to increase the innate immune response to the antigen within APCs). The oligonucleotides, in turn, are coupled to model peptides derived from tumor-associated antigens via specific linker chemistries. Initial studies assessed different linker chemistries and determined their effect on TLR9 activation by the oligonucleotide and antigenicity of the attached peptide. Using a peptide derived from the melanoma gp100 antigen, Skakuj et al. generated SNAs using a liposomal core with TLR9-stimulatory CpG oligonucleotides, immobilized on the core surface through intercalation by using a cholesterol anchor and linked to a peptide derived from the gp100 model antigen [64]. Three different linker chemistries for attaching the peptide to the CpG oligonucleotide were explored and evaluated for their effect on the activation and proliferation of both APCs and CD8^+^ effector T cells: a non-cleavable linker (N-(β-maleimidopropyloxy) succinimide ester, BMPS); a cleavable linker (succinimidyl 3-(2-pyridyldithio)propionate, SPDP), which upon cleavage under reducing conditions within the endosome, leaves a molecular pendant group (3-mercaptopropionamide) attached to the antigen; and a traceless linker (4-nitrophenyl 2-(2-pyridyldithio)ethyl carbonate, NDEC), which upon reduction, engages in intramolecular cyclization followed by the release of the antigen in an unmodified form. All SNA architectures robustly co-delivered the adjuvant and antigen to APCs and effectively activated the APCs, suggesting that the linker chemistry does not impact SNA uptake endosomal TLR9 activation within APCs. Using T cell receptor transgenic CD8^+^ T cells that specifically recognize gp100, SNAs with a traceless linker most effectively induced effector T cell expansion and activation by promoting IFN-*γ*, TNF-*α*, granzyme-B, and IL-6 production [64]. Further optimization of the reduction-labile traceless linker showed that accelerating the rates of chemical dissociation of antigens from CpG oligonucleotide–peptide conjugates resulted in more potent immunostimulatory SNAs [65]. Taken together, these proof-of-concept studies highlight the power of SNA-enabled vaccines to co-deliver the adjuvant and antigen to immune cells, and to more potently enhance APC activation and the priming of antigen-specific T cells compared to linear cues. These data also show that the choice of linker chemistry impacts peptide antigenicity and T cell activation and expansion and showcase the importance of adjuvant/antigen structural arrangements for eliciting potent immune responses.

### 3.3. The Concept of Rational SNA Vaccinology

As a modular and chemically well-defined structure assembled from synthetic components, SNA-based vaccines allow for the systematic variation of the vaccine architecture to optimize the timing of APC activation through the induction of cell surface co-stimulatory molecules and the intracellular processing of the peptide antigen. Using a ‘rational vaccinology’ approach, Wang et al. [66] began to optimize SNA vaccine performance. This study used three compositionally equivalent but structurally distinct architectures. All architectures consisted of a unilamellar liposomal core 3’cholesterol-conjugated with a radial shell of TLR9 agonistic oligonucleotides but differed in the position of the peptide antigen. ‘Encapsulated’ (E-)SNAs contained the soluble antigen within the liposome core. ‘Anchored’ (A-)SNAs displayed the antigen peptide at the surface via chemical conjugation to shorter oligonucleotide linkers adsorbed to the liposome surface. ‘Hybridized’ (H-)SNAs presented the antigen on a linker oligonucleotide that was hybridized to the CpG oligonucleotides. Ovalbumin-1, melanoma-derived antigen gp100, and human papillomavirus-16 oncoprotein E6 antigen were chosen as model antigens. This study demonstrated that the SNA structure through the adjuvant and antigen’s specific spatial and temporal presentation profoundly influences vaccine performance. An assessment of SNA uptake and intracellular trafficking, APC and T cell activation, and anti-tumor responses in vivo demonstrated the superior activity of H-SNAs, compared to E- and A-SNAs, and a linear adjuvant/antigen mixture. On a cellular and molecular level, H-SNAs more robustly co-delivered CpG oligonucleotide and peptide to dendritic cells and demonstrated optimal kinetics of CpG and antigen processing. In turn, the co-delivery of both immunological cues resulted in the synchronized presentation of both antigen and co-stimulatory markers, optimized dendritic cell-T cell interaction, and most effective T cell responses and in vivo anti-tumor effects [66]. Such a precise architectural control of vaccine components was also highlighted in two recent studies of immunostimulatory SNAs carrying prostate-specific antigens, i.e., prostate-specific antigen (PSA), prostate-specific membrane antigen (PSMA), prostate acidic phosphatase (PAP), and T-cell receptor γ alternate reading frame protein (TARP) [67,68]. These prostate cancer-specific SNA vaccines more effectively activated cytotoxic T cells and induced effector memory than the linear mixture of adjuvant and peptide antigen, suggesting that antigenic peptides with previously reported suboptimal activity could be reconfigured into a structurally optimized SNA vaccine for more potent anti-tumor effects [68].

### 3.4. Tumor-Lysate Loaded SNAs as Cancer Vaccines

Vaccines using specific (neo-)antigen-derived peptides represent a rational approach for tumor types, which ubiquitously and homogenously express known tumor-associated antigens. For tumors with less well-defined and heterogeneously expressed antigens, the use of cell lysates derived from a patient’s tumor represents an alternative approach that addresses some of the limitations of using a finite set of well-defined antigens. These include restricted epitope expression by only one of the major histocompatibility complexes (MHCs), inter- and intratumoral heterogeneous expression of antigen, and loss of antigen expression during tumor progression or recurrence. However, the suboptimal cell uptake and bioavailability of tumor cell lysates have limited therapeutic efficacy [69]. To enhance the immunogenicity of tumor cell lysates, Callmann et al. [70] developed liposomal SNA architectures that were prepared from 1,2-dioleoyl-*sn*-glycero-3-phosphocholine (DOPC), loaded with tumor cell lysates derived from triple-negative breast cancer (TNBC) cell lines, and decorated with a shell of radially oriented TLR9-agonistic CpG oligonucleotides (Figure 3). Immune activation was compared to analogous SNA constructs that contained hypochlorous acid (HOCl)-oxidized tumor cell lysates, known to have increased antigenicity in part due to increased proteolytic susceptibility [71] and the presence of aldehyde-modified antigens that are more immunogenic compared to unmodified antigens [72]. As shown for peptide-based SNA vaccines, tumor cell lysate-loaded SNAs enhanced the co-delivery of the adjuvant and antigen to immune cells and extended the survival of syngeneic orthotopic TNBC-bearing mice more robustly compared to non-SNA conjugated mixtures of lysates and oligonucleotide. Notably, oxidized TNBC lysates increased anti-tumor effects of SNAs through enhanced activation of dendritic cells, more profound activation of CD8^+^ T cells, reduced myeloid-derived suppressor cell abundance in the tumor microenvironment, and the induction of long-term immunological memory [70]. These results demonstrate that immunogenicity and the anti-tumor effect of oxidized tumor cell lysates can be enhanced when lysates are presented to antigen-presenting cells in an SNA format. Pending comprehensive assessment of toxicity, tumor lysate-loaded SNAs may represent a potent vaccine platform for cancers that lack known antigenic targets. An initial effort to optimize tumor cell lysate-loaded SNA architectures identified the composition of the liposomal core as a critical design parameter that dictates the overall anti-tumor effect. Here, diacyl lipid tail chain length and degree of saturation of the liposomal core represent critical parameters that influence DNA loading, SNA cellular uptake, serum stability, in vitro immunostimulatory activity, and in vivo biodistribution and anti-tumor activity [73].

### 3.5. High-Throughput Screening to Define SNA Structure-Activity Relationships

The seminal studies described above evaluated the impact of a finite number of structural features on SNA vaccine performance. Now, high-throughput methods for synthesizing and evaluating SNA vaccines combined with machine learning approaches allow for the systematic, large-scale screening of structural features that contribute to in vitro potency. Yamankurt et al. [74] developed a self-assembled monolayers for matrix-assisted laser desorption/ionization mass spectrometry (SAMDI-MS)-based high-throughput assay for the rapid and quantitative measurement of cellular responses to TLR9-agonistic SNAs. RAW-Blue^TM^ macrophages were engineered to secrete NF-κB-inducible embryonic alkaline phosphatase (SEAP). NF-κB is a major transcription factor that is activated by innate immune sensing involving TLRs and other DNA sensors [75,76]. Macrophages were treated with a library of 960 distinct SNA constructs. SEAP-containing media was supplemented with a phosphorylated peptide substrate, and subsequently, both the substrate and dephosphorylated product were captured on monolayers and then analyzed by SAMDI. SAMDI-MS, pioneered by Milan Mrksich, represents a label-free assay for high-throughput, quantitative analysis of enzymatic activity [77,78,79]. SAMDI-MS uses monolayers that present a selective capture chemistry against a background of non-binding tri(ethylene glycol) groups. The monolayers can recognize substrates and products from a complex mixture. A subsequent analysis of the monolayers by MALDI-MS allows for the quantification of the substrate and product, directly correlating with SEAP enzyme concentration. This technology was chosen for its ability to quantify enzyme activities at a high throughput, without dependence on standard optical methods. SAMDI-MS avoids artifacts caused by light scattering and absorbance of the nanoparticles, which are difficult to correct because they are influenced by nanoparticle properties such as size, concentration, and aggregation.

An analysis of SNA structure-activity relationships provided insights into the importance of individual SNA design components and how combinations of these components impact immune activation [74]. The immunostimulatory SNA conjugates chosen for this study consisted of liposomal nanoparticle cores synthesized from DOPC and 1,2-dioleoyl-*sn*-glycero-3-phosphoethanolamine (DOPE), CpG DNA oligonucleotides, and a peptide derived from ovalbumin as a model antigen. Results from this screen revealed that the lipid composition of the core and surface chemistries for oligonucleotide anchoring to liposomal cores, together with oligonucleotide concentration, orientation, and backbone, influenced SNA immune-stimulatory activity [74]. Multifactor analysis of variance, together with the use of machine learning models, revealed that the oligonucleotide concentration heavily influenced SEAP secretion. Second to oligonucleotide concentration, the lipid moiety conjugated to the oligonucleotide for liposome core attachment had the most significant impact on immune activation. In addition, the conjugation terminus of the oligonucleotide also impacted immune activation, as 5′-conjugated SNAs showed significantly higher activity than 3′-conjugated SNAs, particularly for SNAs with cholesterol conjugation. The oligonucleotide backbone also influenced the immune stimulatory activity [80], as SNAs with phosphorothioate backbones generally outperformed their phosphodiester counterparts. Subsequently, empirical HTS data were used to train a non-linear machine-learning algorithm to predict immune activity from SNA properties automatically. In sum, this study, together with other approaches [81,82], indicate the need to consider the full range of structure-activity relationships when designing SNAs and other nanomedicines by high-throughput processes, as nanoconjugate properties can be strongly interrelated in non-obvious ways. In addition, the combination of experimental and computational methods to define and predict SNA immune activity will be instrumental in future efforts to optimize SNA chemistry when consideration is given to more complex architectures, combining multiple adjuvant oligonucleotides and peptide antigens.

## 4. Other Multimodal SNA Architectures in Preclinical Development

In addition to SNA vaccines, the most well-defined class of multimodal SNA nanotherapeutics, a series of SNA architectures have been synthesized and evaluated preclinically. In addition to oligonucleotide payloads, these SNA conjugates co-deliver chemotherapeutic drugs [83,84], antibodies for tumor cell targeting [85], or imaging agents [14,86].

Platinum (Pt) compounds, such as cisplatin or carboplatin, effectively treat a broad spectrum of cancer types, including testicular and ovarian cancers [87,88]. To determine whether Pt compounds, when conjugated to oligonucleotide-functionalized gold nanoparticle cores, are more effective than the free drug, Mirkin, Lippard, and colleagues developed SNAs functionalized with dodecyl amine-terminated DNA oligonucleotides, which were linked to the Pt (IV) pro-drug via amide linkages [83]. Oligonucleotide to Pt (IV) linkage did not affect cell uptake mediated by the oligonucleotide moiety. It also did not alter the reduction potential of the Pt (IV) warhead, as axial ligands of the Pt (IV) complex were removed upon cell entry, resulting in the formation of the active Pt (III) drug. Efficacy testing in human cancer cell lines revealed that SNA-bound Pt (IV) showed a greater than 10-fold reduction in IC_50_ values than the free drug. These data provided the first proof-of-concept that SNAs can be used as delivery vehicles for chemotherapeutic drugs to enhance efficacy and reduce adverse side effects [83]. A similar strategy was used to synthesize and characterize paclitaxel-loaded SNAs. This study aimed to increase paclitaxel aqueous solubility, decrease effective drug concentrations required for maximum cell killing, and reduce adverse side effects associated with drug administration [89]. Here, gold nanoparticle templates were functionalized with amine-terminated DNA oligonucleotides, after conjugating the oligonucleotide to a paclitaxel carboxylic acid derivative via carbodiimide linker chemistry. SNA-bound paclitaxel showed a greater than 50-fold enhancement of aqueous solubility, robust cell uptake, and apoptosis-inducing activity [89].

In addition to increasing chemotherapeutic drug solubilities and efficacies, SNAs can deliver bioactivated contrast agents, including paramagnetic Gd^III^ complexes. Song et al. [86] developed Gd^III^-conjugated SNAs as an MRI agent, enabling high-level Gd^III^ loading, efficient cellular uptake, and excellent contrast enhancement. Gd^III^ was conjugated to poly DNA thymine (poly dT) oligonucleotides and linked to gold nanoparticle cores via thiol adsorption using copper-catalyzed click chemistry. Gd^III^-SNAs showed high-level cell uptake and brighter MRI signal compared to non-SNA-conjugated Gd^III^. When intratumorally administered to glioblastoma-bearing mice, these architectures were used to define SNA pervasive infiltration of intracranial tumors using ex vivo MRI on resected tumor-bearing mouse brain [14]. In sum, these data demonstrate that SNAs can be used as a cell and tissue-permeable, biocompatible MR contrast agent with high Gd^III^ loading, relaxivity, and cell uptake.

## 5. Conclusions and Outlook

SNAs architectures, i.e., nanoparticle core templates densely packed with highly oriented oligonucleotides, represent emerging and potentially paradigm-shifting oligonucleotide-based therapeutics. Comprehensive preclinical characterizations revealed that SNAs are recognized and endocytosed by scavenger receptors; potently activate endosomal TLRs; deliver peptides to antigen-presenting cells; and following endosomal escape, can potently and persistently neutralize gene expression due to increased resistance toward nuclease-driven degradation. These preclinical studies have now been translated into the clinic, where early phase clinical trials demonstrate efficacy of gene-regulatory and immunostimulatory SNAs in solid tumors.

What is next? The development of high-throughput screening and machine learning approaches to investigate SNA structure-activity-relationships and define and predict critical design parameters to optimize SNA activity demonstrated that the systematic variations in SNA structure can lead to significant improvements in SNA performance. Enabled by their modular synthesis and chemically well-defined building blocks, SNAs will allow for the development of potent gene-regulatory conjugates or vaccines with well-defined presentation and stoichiometry of components.

For vaccine development, SNAs must be systematically innovated for optimal antigen-specific T cell activation by incorporating multiple peptide antigens that engage both MHC-I and MHC-II, by improving tumor cell lysate loading into hollow nanoparticle cores, by fine-tuning SNA surface and linker chemistries for the most robust co-delivery of immunostimulatory cues to antigen-presenting cells, and by developing oligonucleotides that target multiple innate immune receptors or intracellular DNA sensors. In addition to developing the next generation of SNA vaccines for clinical development, this line of research will also help us better understand the structural and mechanistic basis for vaccine function through systematic iterations in SNA synthesis and design (Figure 4).

As concerns the optimization of gene-regulatory SNAs, we must understand, on a fundamental level, how SNAs architectures can escape the endosome upon SR-A-dependent internalization, how to increase cell uptake and cytosolic localization, and augment siRNA loading on nanoparticle templates to engage the RNAi machinery more effectively. A recent study exemplified the need to further optimize gene-regulatory SNAs. Vasher et al. [90] developed a novel SNA architecture in which both strands of a hairpin-like siRNA oligonucleotide were attached to the gold core template to prevent guide strand dissociation from the passenger strand. Such strand dissociation is typically observed with first-generation SNAs, in which the passenger strand is attached to the gold core via thiol adsorption, and the guide strand is linked to the passenger strand via complementary base pair hybridization. This next-generation gene-regulatory SNA construct achieved higher loading and increased the delivery of active siRNA duplexes, longer siRNA half-life, and more durable gene knockdown [90]. Another design parameter to consider is the presence of guanine-rich sequence motifs. Guanine-rich elements can form G-quadruplexes, which present a negative charge at a higher density due to the unique secondary structure and folding, and as discussed above, result in more effective engagement of SR-As and cell uptake [91,92,93].

Furthermore, we must begin to develop innovative treatment regimens that combine the power of SNA-enabled gene regulation to neutralize virtually any oncogenic lesion and the proven efficacy of chemo- and targeted therapeutics. The concept is to design and comprehensively characterize multimodal SNA architectures carrying both chemotherapeutic drugs and gene-regulatory oligonucleotides to target critical factors that drive chemo- and targeted therapy resistance (Figure 4).

Besides optimizing the oligonucleotide shell for the activation of innate immunity, vaccine development, and gene regulation, future studies must focus on the nanoparticle template as a therapeutic cue. For example, gold nanoparticle cores were shown to act as a radiosensitizers [94]. Radiosensitization is based upon the phenomenon that X-rays are absorbed by gold-based nanostructures, resulting in the release of low-energy electrons from the nanoparticle template, and the deposition of energy in water in the form of radicals and electrons. Hydroxyl radicals form, which in turn promote DNA strand breaks and enhance radiation-induced tumor cell death [94]. Advanced testing in preclinical mouse models and in clinical trials will determine whether gold-based SNA architectures can be used as adjuvant to radiation therapy, particularly when carrying gene regulatory oligonucleotides targeted to radioresistance factors.

Recent studies have shown that proteins can be used as SNA templates, as the ability of SNAs to enter cells and tissue is mediated by the unique 3D architecture of the oligonucleotide shell and is independent of the type of core material. By densely functionalizing a protein’s surface with a shell of radially oriented oligonucleotides, protein-based SNAs (ProSNAs) can delivery cell-impermeable bioactive proteins to cells and tissue, particularly when conjugated with G-quadruplex-containing oligonucleotides [95,96,97]. Such an approach can be used to replace dysfunctional proteins, or to inhibit oncogenic signaling processes, e.g., via the delivery of dominant negative mini-proteins. ProSNAs now await preclinical testing in murine cancer models.

Taken together, these future preclinical and clinical studies are poised to establish gene-regulatory and immunostimulatory SNAs as a customizable, on-demand therapeutic platform that can rapidly be developed to combat a broad spectrum of cancer types.

## 6. Patents

A.H.S. is the inventor on patent US20150031745A1, which describes SNA nanoconjugates to cross the blood–brain barrier.

## Figures and Tables

**Figure 1 cancers-14-01615-f001:**
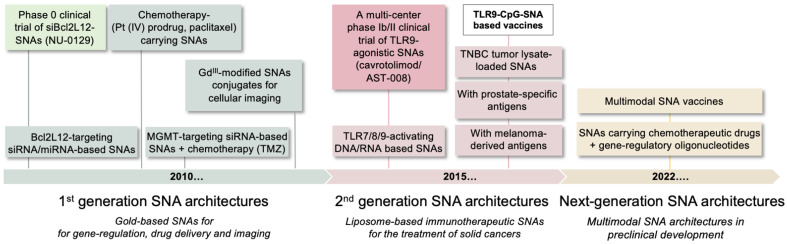
Timeline of SNA preclinical and clinical development. First-generation SNA architectures focused on gene-regulation and the delivery of chemotherapeutic and imaging agents. Second generation SNAs are liposomal structures designed to active anti-tumor immune responses. Next-generation SNAs are multimodal structures, functionalized with multiple therapeutic cues. Bcl2L12, Bcl2Like12; TMZ, temozolomide; TLR, Toll-like receptor; TNBC, triple negative breast cancer.

**Figure 2 cancers-14-01615-f002:**
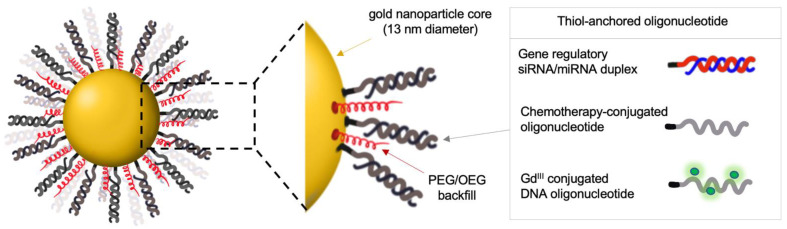
Gold-based SNAs for gene regulation and delivery of chemotherapeutic and imaging agents. OEG, oligo (ethylene glycol); PEG, poly (ethylene glycol); Gd^III^, gadolinium III; siRNA, small interfering RNA; miRNA, microRNA.

**Figure 3 cancers-14-01615-f003:**
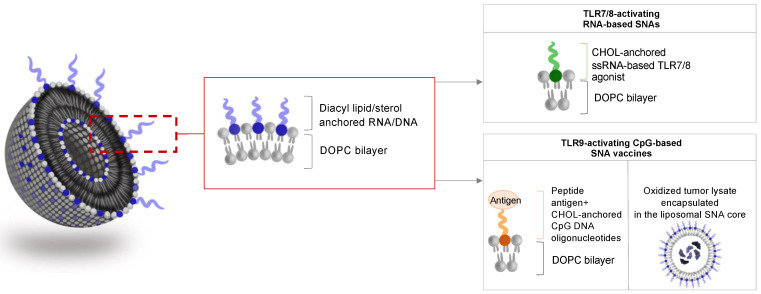
Liposomal SNAs for immune activation. SNAs are designed to carry TLR-activating oligonucleotides and antigenic peptides or are loaded with tumor cell lysates. DOPC, 1,2-dioleoyl-sn-glycero-3-phosphocholine; CHOL, cholesterol.

**Figure 4 cancers-14-01615-f004:**
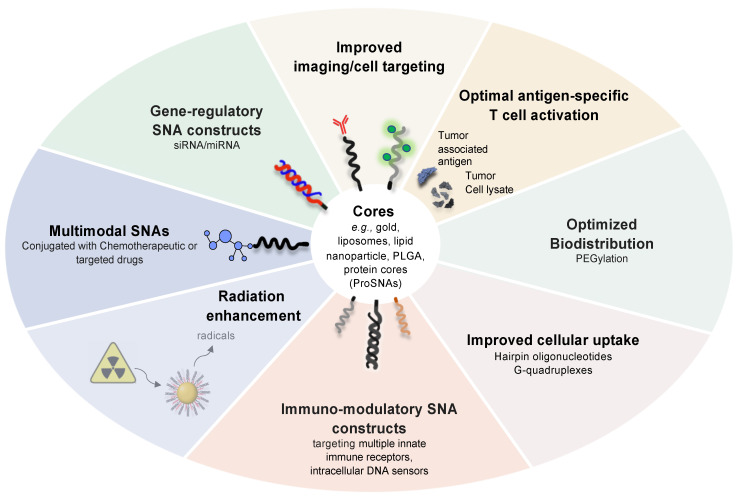
Next-generation SNA architectures for gene regulation and immunostimulation.

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
