# Peer review of "Spherical Nucleic Acids as Precision Therapeutics for the Treatment of Cancer—From Bench to Bedside"

_cancers, 2022, doi:10.3390/cancers14071615_

Round 1

Reviewer 1 Report

The review by Mahajan and Stegh highlights recent advances in anticancer therapeutic application, such as gene silencing or immunomodulation, of spherical nucleic acids (SNAs), which comprise nanoparticulate core, e.g. metal (Au, Ag), oxide (Fe3O4, SiO2), polymer (PLGA), or lipid nanoparticle, covered with a dense shell of oligonucleotides, such as DNA, RNA, LNA etc. The modular design of SNAs greatly facilitates their development into multifaceted therapeutic tools that arguably surpass 'regular' oligonucleotides in terms of cellular and tissue uptake, and biological activity. The paper is timely and well-written, and, to the opinion of this Reviewer, deserves rapid publication after a very minor revision.

I spotted a few typos, which need correction. My suggestions are given below.

Line 54 - Functionalized.

Line 91 - Preclinical.

Lines 104-105 - Better to say: "... antisense oligonucleotides (ASOs) or small interfering RNAs (siRNAs)".

Line 117 - Similarly to the above, "... microRNAs (miRNAs)".

Line 172 - Please check Fig. 2 that text labels do not overlap with the pictures.

Line 205 - Please re-phrase, e.g. "... delivery of siRNA therapeutics".

Lines 268-270, Fig. 3 - Please replace the abbreviation CHOL for cholesterol with more conventional Chol.

Line 313 - "... SNA uptake OR endosomal TLR9 activation".

Line 553 - Preclinical.

Author Response

We thank the reviewer for his positive assessment of our review. All corrections were made as suggested, as indicated in the revised manuscript. We also revised Figure 2, and corrected one typo in Figure 1.

Reviewer 2 Report

The manuscript "Development and Investigation of Nanoconstructs Suitable for Cancer Therapy and Diagnosis" by Alexander H. Stegh and Akanksha S. Mahajan describes current achievements in SNA applications for cancer therapeutics. Authors thoroughly analyzed the field: briefly describe development of SNA construction and focus on preclinical aspects. The review is well written and is valuable both for people in cancer and nucleic acid therapeutics. Outlook is really great - discussion of probable future developments is valuable for readers.

I suggest to decrease the degree of SNA superiority in vaccines (at least). SNA vaccinology is a bit strange term as all oligos can be used only as adjuvants, being not the main players. Extraction of total nucleic acids from tumors followed by encapsulation in liposomes seems to be not very perspective approach in clinic due to safety concerns. And in this specific case introduction of oligo agonist of TLR9 does not make this system SNA (if we evaluate a number of oligos per liposome). 

I acknowledge the work being performed in this detailed study, but due to issues described above I suggest resubmitting the manuscript after minor revision. 

Author Response

We thank the reviewer for his positive assessment of our review.

  1. We highlighted potential safety concerns associated with tumor lysate-bearing SNAs (line 399-401).
  2. We would like to emphasize that in preclinical studies, SNA vaccines are indeed superior to linear mixtures (free DNA and antigen ) (e.g., Wang et al., PNAS 2019).
  3. SNA vaccines are not only functionalized with adjuvant oligonucleotide, but also with one or multiple peptide antigens, or alternatively, with tumor cell lysates. The latter architectures are useful for tumors with unknown or less well-defined antigen landscape. These lysates do not contain nucleic acids, but instead, are enriched for proteins and lipids. Given the multi-component nature of SNA vaccines, the term ‘vaccinology’ is adequate.
  4. Structures that consists of a nanoparticle core mono-functionalized with TLR9 oligonucleotide should be referred to as 'SNAs'. Per definition, SNAs are structures that consists of a nanoparticle core conjugated with a corona of nucleic acids.